# Why the MARS2 Trial Does Not Mean the End of All Mesothelioma Surgery

**DOI:** 10.3390/cancers17050724

**Published:** 2025-02-21

**Authors:** David Waller, Rocco Bilancia, Luigi Ventura, Sara Tenconi, Laura Socci, Andrea Bille

**Affiliations:** 1St Bartholomew’s Hospital, London EC1A 7BE, UK; 2Golden Jubilee National Hospital, Glasgow G81 4DY, UK; rocco.bilancia@gjnh.scot.nhs.uk; 3Northern General Hospital, Sheffield S5 7AU, UK; luigi.ventura@nhs.net (L.V.); sara.tenconi@mft.nhs.uk (S.T.); laura.socci@uhbw.nhs.uk (L.S.); 4Guy’s Hospital, London SE1 9RT, UK; andrea.bille@gstt.nhs.uk

**Keywords:** pleural mesothelioma, pleurectomy/decortication, surgical trials

## Abstract

In order to challenge the published conclusions of the MARS2 trial, we have performed a post hoc analysis of a representative sample of 50% of the surgical arm of the trial. We have re-analysed their selection criteria based on contemporary staging and pathological requirements. We have found that only 1 in 3 of this group would be offered surgery in current practice. Their survival has also been found to be significantly higher than in the main trial cohort. It is our contention that these findings raise sufficient doubt about the validity of the MARS2 conclusions to justify our belief that this trial does NOT mean the end of all mesothelioma surgery.

## 1. Introduction

Following the publication of the results of the MARS2 trial [1], there has been further new controversy regarding the role of radical surgery in the management of pleural mesothelioma. Many surgeons, oncologists and other physicians have concluded that the results mean that radical surgery should be consigned to history [2] whilst many surgeons in the field believe that the trial has many flaws which detract from the far-reaching conclusions [3]. These criticisms revolve around deficiency in patient selection, the choice of an idiosyncratic chemotherapy scheme and relatively high early mortality (9%) with poor outcomes in the surgical cohort.

As the lead recruiter in the trial, I feel justified in admitting, on behalf of my co-authors, that the problems associated with MARS2 relate to the following factors. We operated on too many patients with known poor prognostic factors, i.e., both histological and physiological and non-response to chemotherapy. We operated too late in the course of disease on many with locally advanced tumours. Consequently, we operated by taking too much tissue, particularly the diaphragm.

The latest 9th TNM staging for mesothelioma [4] has modified the T component based on the sum of CT measurements of maximum pleural thickness at 3 levels and a measurement of the maximum thickness of the interlobar fissure. Using this latest TNM staging, the median survival for the pathological pstage I (T1N0) is 50 months and for pstage II (T1N1 or T2N0) is 29 months, with 40% alive at 3 years. Furthermore, in the 9th TNM edition, patients with pN0 pathologically node negative had a median survival of over 33 months. Based on these results, which are almost exclusively from patients undergoing surgery as they are based on pTNM, there does appear to be a reasonable case for considering surgical resection in these patients.

The purpose of our review was to re-examine some of the results and therefore challenge the conclusions of the MARS2 trial.

As part of our critique, we aimed to re-evaluate the trial findings by applying contemporary selection criteria [5] to a representative sample of the trial’s surgical cohort. We made the assumption that in modern, specialist mesothelioma centres, resection would be limited to those with epithelioid stage I or II disease. Therefore, we aimed to estimate how many who were randomised to the surgical arm of the trial would proceed to surgery in current practice.

Importantly, we intended to challenge the assertion made in the MARS2 manuscript that all mesothelioma should be viewed as unresectable including early-stage epithelioid disease. We specifically wished to challenge the conclusion that, therefore, all patients (irrespective of clinical stage) should receive immunotherapy. Particularly as this treatment may have more risks than benefits in the population most suitable for surgical resection.

## 2. Materials and Methods

We retrospectively analysed the selection and outcome of the patients in the surgical arm of the MARS2 study who were operated upon only by the authors in 4 of the 5 trial surgical centres: St Bartholomew’s, Golden Jubilee Glasgow, Northern General Sheffield and Guy’s. These were the only data made available to us on request.

We revised the clinical staging of these patients by applying the criteria in the 9th TNM edition [4].

We did this by re-analysing, with the assistance of the local radiologist, the pre-treatment axial CT scans. As defined in [4], the chest was divided into three compartments with a virtual demarcation at the level of the top of the arch of the aorta and the first image at or below the level of the left atrium, dividing the chest into three equal parts. The maximum pleural thickness measurements were taken perpendicular to the chest wall or mediastinum in the area of maximal pleural thickness in each third of the hemithorax (pmax1, pmax2 and pmax3) and combined to estimate the sum of the maximal pleural thickness at the three levels (Psum = pmax1 + pmax2 + pmax3). Maximal pleural thickness was also measured along the fissures (Fmax) We also noted the clinical nodal status as either N0 or N1 depending on a maximum transverse diameter of 10 mm in the visible nodes in the ipsilateral hemithorax. We subsequently analysed post-resection histology reports to record those patients who were found to have nodal metastases in mediastinal stations. Finally, overall postoperative survival was recorded from retrospective analysis of patient records.

We applied the following exclusion criteria that we would assume would preclude surgery in the contemporary era: those with non-epithelioid mesothelioma on pretreatment biopsy; those with presumed epithelioid mesothelioma who would be reclassified as clinical stage III disease in the 9th TNM revision either due to local mediastinal or chest wall involvement or with Psum > 30 mm; those with clinical nodal N1 disease and Psum > 12 mm (T2N1) and those found to have pN1 disease that, in experienced centres, would have been accessible to pre-resection biopsy by endobronchial ultrasound (EBUS) or cervical mediastinoscopy in stations 2, 4 or 7. We then extrapolated these results to estimate what proportion of the MARS2 surgical cohort would actually have been considered in current practice.

Intergroup differences were analysed using a Mann–Whitney U-test for non-parametric data. Statistical significance was accepted for a *p* value < 0.05.

## 3. Results

In the MARS2 trial, a total of 158 patients underwent surgery [1]. We analysed the records of 79 patients (50% of this population) who had undergone resection in four of the five trial surgery centres: St Bartholomew’s 59 patients, Glasgow’s 10 patients, Sheffield’s 6 patients and Guy’s 4 patients. Data were not available from one trial centre at Glenfield Hospital, Leicester and from one trial surgeon in Sheffield.

The clinical stages of the disease as submitted to the MARS2 trial are shown in Table 1.

This cTNM staging refers to the 8th TNM staging system [6]. The locally advanced cT3 cases referred mainly to mediastinal pleural involvement or non-transmural pericardial invasion with isolated cases of focal (resectable) chest wall invasion. The two cT4 cases referred to apparent transmural pericardial invasion.

Following recalculation of clinical staging using the criteria in the 9th TNM revision, we divided the 79 patients into two groups. In Group A, 52 patients were found to have a clinical stage of T2N1 or above or to have non-epithelioid mesothelioma. Only 27 (34%) patients remained in epithelioid stage I or II.

### 3.1. Reasons for Exclusion in Group A (Table 2)

#### 3.1.1. Histological Sub-Type

We excluded five patients on the basis of non-epithelioid mesothelioma in the pre-treatment biopsies, with four biphasic and one pure sarcomatoid.

**Table 2 cancers-17-00724-t002:** Exclusion characteristics—group A.

Trial Centre	1	2	3	4	Total
Non-epithelioid	1	2	0	2	5
cT3/4	25	4	1	2	32
Psum > 30 mm	3	0	0	0	3
cT2N1	9	1	0	0	10
cT2 pN1	2	0	0	0	2

#### 3.1.2. Locally Advanced Disease

Clinical stage T3 or T4 disease was reported in 32 of the 55 patients in group A. This was predominantly due to reported invasion of the mediastinal pleura.

Using measurements of maximal pleural thickness (Pmax) at the specified three levels, the sum of the thicknesses (Psum) was greater than 30 mm in three further patients who had been previously reported to have cT1 or T2 disease with no local invasion.

#### 3.1.3. Clinical Nodal Involvement

On the pre-treatment CT scans, clinically involved ipsilateral lymph nodes were reported as enlarged (cN1) in 10 patients with cT2 tumours.

#### 3.1.4. Pathological Nodal Involvement

Analysis of the post resection histology (Table 3) revealed pathological nodal metastases in stations 4, 7 or 10, which would have been amenable to biopsy and histological confirmation by endobronchial ultrasound or cervical mediastinoscopy in a further two patients who were originally classified in c stage T2N0 epithelioid. We assume they would therefore have been classified as cT2N1 and excluded.

### 3.2. Reasons for Inclusion in Group B (Table 4)

#### 3.2.1. Stage I Disease

Clinical stage I (T1N0) disease based on the criteria of the 9th TNM Edition with a Psum < 12 mm was found in 12 of the 79 (15%) patients.

**Table 4 cancers-17-00724-t004:** Inclusion characteristics—group B.

Trial Centre	1	2	3	4	Total
cT1N0	8	2	2	0	12
cT2N0	4	1	1	2	8
cT1N1	1	0	0	0	1
cT1 pN1	6	0	0	0	6

#### 3.2.2. Stage II Disease

Clinical stage II (T1N1 or T2N0) disease based on the 9th TNM Edition radiological assessment was found in nine (11%) patients. These included one patient who was upstaged from T1 to T2 on the basis of Psum > 12 mm.

#### 3.2.3. Pathological Nodal Involvement

Clinical stage II (T1N1) was found in a further 6 of the 79 (8%) patients whose postoperative pathology revealed nodal metastases which reasonably could have been accessed by EBUS or mediastinoscopy in stations 2, 4 or 7.

### 3.3. Postoperative Results

#### 3.3.1. Postoperative Pathology

On postoperative pathological analysis of the resected specimens, epithelioid mesothelioma was reported in only 22 of the 79 (28%) patients. Of these, only 12 of 79 (15%) remained in epithelioid stage I or II (Table 3).

#### 3.3.2. Survival

There was a significant intergroup difference in overall survival: Group A: 8.5 (1–55) months vs. Group B 32 (1–72) months *p* < 0.005 (Mann–Whitney U test).

## 4. Discussion

The conclusions of the MARS2 trial had far-reaching implications. The reported finding led to the assertion that all mesothelioma cases should be considered as unresectable [1]. This controversial statement was based on the crude assertion that radical surgery plus chemotherapy had no survival benefit over platinum/pemetrexed chemotherapy alone and indeed may have been detrimental. This was based on the findings that the addition of extended pleurectomy/decortication to chemotherapy reduced overall survival from 24.8 months to 19.3 months. [1]. It was additionally suggested that surgery resulted in more serious side effects, poorer quality of life and inferior cost-effectiveness. However, the trial protocol has, in retrospect, many shortcomings that question its application to contemporary practice.

Our findings in this study, in what we believe to be a representative sample of the trial population, illustrate the problem in extrapolating the results of MARS2 to the whole practice of mesothelioma surgery. We suggest that only a small proportion (34%) of the MARS2 surgical population would satisfy contemporary selection criteria. Unsurprisingly, this group was found to have significantly greater survival than those who were excluded in our study or the whole group who underwent surgery in the main MARS2 trial. Thus, contrary to the conclusions regarding early-stage disease by Lim in the main trial paper [1], we contend that this eventual small number would invalidate any statistical comparison of the true stage I/II disease patients.

The main criticism would be that the MARS2 selection criteria were too liberal and were intentionally pragmatic to ensure adequate recruitment. It may be argued that within MARS2, the subgroup of epithelioid T1-2N0 patients showed no benefit for surgery over no surgery [2]. However, our analysis shows the flaws in this comparison since so many were under staged.

In the planning of MARS2, we were conscious of the problems in recruitment encountered in the initial MARS trial [7], and, therefore, we were more pragmatic and liberal in the inclusion criteria. We assumed that randomisation would equate parameters including nodal status and cell-type. Whilst these assumptions allowed us to make a valid comparison, the effect on the eventual application to clinical practice were not considered sufficiently. Nevertheless, we recruited to MARS2 ahead of schedule even through the COVID-19 pandemic, but there were consequences. As many were included with poor prognostic factors, the errors in the selection criteria became clear. We had enrolled too many subjects.

Too many were recruited who had non-epithelioid mesothelioma on diagnostic biopsy, as we have found in our sample, despite the knowledge that this cell sub-type was known to be associated with a poor prognosis, one which did not seem to be influenced by surgical treatment [8,9]. In the MARS2 trial, one in seven patients had this poorer prognosis type. Furthermore, with a mean age of 69 years, many of the subjects were elderly and had a mean FEV1 of 75%, so it could also be argued that too many elderly patients with at least moderate COPD were subjected to extended pleurectomy/decortication.

The simplistic surgical objective was to obtain a macroscopic complete resection [10], irrespective of nodal disease, so long as the nodes were confined to the ipsilateral hemithorax. This objective was achieved in an impressive 84% of cases despite the advanced stage of many tumours. Much of the selection in the trial was based on the 8th TNM Edition of staging [6], and in this context, 56% had locally advanced disease with pericardial and mediastinal involvement. PET-CT was not mandated nor was histological confirmation of mediastinal nodal status. Thus, as we found in our sample, many had nodal metastases in otherwise resectable disease, and, furthermore, in the main trial, 4% of surgical patients had peridiaphragmatic nodes classified as cM1 disease. In our sample, the proportion with disease in stage III in the 9th TNM edition [4] with either locally advanced or high volume (Psum > 30 mm) disease amounted to nearly two thirds.

As a consequence of being required to resect locally advanced disease, it can be argued that too much tissue was removed. Extended pleurectomy decortication, which includes diaphragm resection, was performed in 83% even in those with no pathological involvement of the diaphragm. Diaphragm preservation in the procedure of pleurectomy/decortication [11] has been shown to result in less peritoneal progression and improved survival over diaphragm resection and reconstruction [12].

The physiological deficit from losing the function of the hemidiaphragm [13] may explain the excessive treatment-related mortality of 12% in the MARS2 study related to pneumonia in most cases. It should also be noted that the excess in mortality in the surgical arm of the trial compared to the no-surgery arm (12% vs. 6%) was not disease related. Indeed, there was no difference in the disease progression in the two groups.

It can also be suggested as a reason why those in the surgical arm had a transient reduction in health status. As would be expected from a major thoracotomy, the reported quality of life dropped at 6 weeks but then recovered to similar levels to those in the no-surgery arm. It must also be noted that full quality of life data were only obtained on around half of the trial participants.

A further inevitable consequence of thoracotomy was the increased treatment-related side effects. Closer analysis reveals that the major differences between surgery and no surgery were due to atrial fibrillation, a recognised transient complication of thoracic surgery [14], and persistent air leak, which one would hope would not be seen in a patient receiving chemotherapy alone.

The extent of surgical resection may also have been contributory to the significant difference in receipt of further postoperative systemic therapy (either chemotherapy or immunotherapy) in MARS2. Although, one should acknowledge that the trial design favoured the non-surgery arm. Those who progressed on the first two cycles of chemotherapy did not continue in the non-surgery arm, but, providing the disease remained resectable, patients in the surgery arm continued to operation. Having survived the operation, their oncologists were reluctant to give further chemotherapy as the initial two cycles had been ineffective. The resulting statistic that 40% of the surgery arm received no further systemic therapy and, therefore, did not complete the trial protocol should be borne in mind when interpreting the results. It has been accepted for many years that unimodal surgical treatment is inadequate for mesothelioma [15].

The protocol of giving only two cycles of neoadjuvant chemotherapy was decided upon to minimise the number of participants who would be excluded prior to randomisation due to disease progression. However, this protocol of breaking the chemotherapy regime in order to operate is in contradiction to established oncological principles [16]. This protocol may not only have been insufficient to obtain a reduction in tumour volume but also sufficient to induce a negative effect on tumour biology. Since completing recruitment to MARS2, we have found there to be a detrimental effect from neoadjuvant platinum/pemetrexed chemotherapy in stage I and II disease. We have reported an association with the epithelial-to-mesenchymal transition (EMT) [17] between pre- and postoperative histology and neoadjuvant chemotherapy [18]. Thus, the unexpectedly high incidence in postoperatively detected biphasic mesothelioma (28% in this study) was associated with inferior survival.

We note that the recently published EORTC 1205 study found no survival difference between neoadjuvant or adjuvant chemotherapy and pleurectomy decortication [19]. However, although the trial numbers were relatively small, the selection bias favouring three cycles of neoadjuvant therapy, with the exclusion of non-responders, was not seen. There has not been an impressive benefit from neoadjuvant chemotherapy even up to four cycles [20], and this lesson must be learned for future surgical trials.

Were the outcomes from surgery affected by the inexperience of the surgeons as has been suggested by other non-trial surgeons [3]? No, because there was careful cross-validation of more junior operators during procedures by the most experienced surgeons to achieve quality assurance. Our operative procedure was standardised to remove all suspicious pleural tissue with more extensive visceral pleurectomy than other centres have recommended, including diaphragm resection. Other authors have criticised this apparently aggressive approach [21], but this surgical approach reflects the more advanced disease seen in MARS2 when over 40% in our sample had clinical evidence of diaphragm involvement. Furthermore, the criticism that the 90-day mortality in MARS2 exceeded international benchmarks [22,23,24] also ignores the benefits of selection bias seen in non-randomised case series

There was some confusion in the MARS2 reporting which did not clearly differentiate between trial centres which administered the chemotherapy and the small group of five centres in which the surgery was performed. The chemotherapy regime was commonly administered and, therefore, did not require specialist oncologists, resulting in many centres treating only a handful of patients in the trial. This fact may have given the wrong impression that the operations were performed in inexperienced centres [3]. In fact, each specialist surgical centre was served by several oncology centres.

Clearly, future research into the role of radical surgery for mesothelioma is still indicated. As a result of the misleading interpretation of MARS2, currently the younger, fitter patients who present with stage I or II epithelioid disease in UK face the prospect of expectant observation and symptom control only. Many oncologists would like to extend the administration of dual agent immunotherapy to all-comers. Whereas this regime is licenced in later stage, unresectable disease on the basis of the randomised Checkmate 743 trial [25], there are real-world concerns that the risks of side effects may outweigh benefits in the epithelioid sub-type of mesothelioma [26,27]. There also many biological reasons chemotherapy may be preferred for epithelioid disease [28].

We believe that a MARS3 trial remains a possibility. Following on from the shortcomings of MARS2, careful patient selection and diaphragm-sparing surgery are mandatory. The timing of additional systemic therapy (chemo or immunotherapy) remains to be determined. The lack of suitable patients presenting in early stage will be highlighted as a reason not to proceed, but earlier detection, with wider access to thoracoscopy [29], and early multidisciplinary discussion [30] suggest that patients will be identified at an earlier stage with better performance status.

## 5. Conclusions

Our post hoc analysis of a subgroup of 50% of the MARS2 trial surgical arm suggests only around one in three would have met current surgical selection criteria of stage I or II epithelioid mesothelioma. Consequently, we would argue that the small number of subjects, even if equally distributed in each arm of the study, would be underpowered to justify the final conclusions. Thus, the MARS2 trial cannot be used to justify the termination of all mesothelioma surgery. Its worth will be in emphasising the importance of early, favourable case selection enabling preservation of respiratory function.

## Figures and Tables

**Table 1 cancers-17-00724-t001:** Original cTNM staging as submitted to MARS2.

	T1	T2	T3	T4
N0	22	11	21	0
N1	1	10	12	2

**Table 3 cancers-17-00724-t003:** pTNM staging.

	T1	T2	T3	T4
N0	6	3	29	3
N1	3	4	24	7

## Data Availability

The data presented in this study are available in this article.

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
