# Peer review of "Why the MARS2 Trial Does Not Mean the End of All Mesothelioma Surgery"

_cancers, 2025, doi:10.3390/cancers17050724_

Round 1

Reviewer 1 Report

Comments and Suggestions for Authors

Manuscript ID: cancers-3475387

Why the MARS2 trial does not mean the end of all mesotheli-2 oma surgery

Cancers

Dear Editor,

The authors critically assess the conclusions of the MARS2 trial, which suggested that extended pleurectomy/decortication (EPD) for pleural mesothelioma offers no survival benefit over chemotherapy alone and may even be detrimental. They argue that the trial's findings should not be overgeneralized due to key limitations in patient selection, staging, and surgical execution. A retrospective analysis of a subset of surgical patients from MARS2, re-evaluated with contemporary selection criteria and the 9th TNM staging system, indicates that only a minority would have been deemed suitable for surgery today. These findings challenge the assertion that all mesothelioma cases are unresectable and underscore the need for a more refined patient selection process in future trials. The authors advocate for a re-examination of surgical strategies and propose a new trial focusing on early-stage epithelioid mesothelioma with rigorous criteria to better assess the role of surgery in treatment outcomes​. The fact that the principal investigator of the MARS2 trial has personally acknowledged the study's limitations is a significant development that could bring hope to malignant pleural mesothelioma (MPM) patients worldwide. From the perspective of evidence-based medicine, clarifying the blind spots and weaknesses of the MARS2 trial is of paramount importance. As previously mentioned, criticisms of MARS2 from external observers who were not involved in the trial may risk being dismissed as mere dissenting opinions from surgeons with vested interests. However, testimony from physicians who directly participated in the trial carries substantial weight and has the potential to become a seminal contribution to the literature on MPM treatment. That said, several key aspects regarding research ethics must be explicitly addressed. Only with appropriate revisions and transparent disclosures can this manuscript be deemed suitable for publication in Cancers.

  1. A critical ethical concern regarding this study is whether the reanalysis of MARS2 trial data falls within the scope of the original informed consent. If the initial consent was limited to analyses strictly within the trial framework, additional IRB approval and, if feasible, participant re-consent may be required. However, if the trial protocol explicitly permitted secondary analyses or follow-up investigations, this reanalysis may not pose significant ethical issues. Given the importance of this issue, I recommend that the Editor inquire with the authors regarding the ethical approval status and scope of informed consent for this study. To ensure clarity and transparency, please explicitly describe how these ethical aspects were considered and what procedures were undertaken to comply with ethical guidelines.
  2. A key issue in this study is the verifiability of its conclusions. The authors argue that appropriate patient selection could demonstrate the effectiveness of surgery; however, verifying this claim would likely require a new prospective trial rather than a retrospective reanalysis of selected MARS2 cases. Since retrospective analyses inherently limit the ability to establish causality, it may be more appropriate to position this study as a hypothesis-generating investigation rather than one that provides definitive evidence. While presenting survival data using the 9th TNM staging is valuable, it should be explicitly stated that these findings serve as a basis for future trial designs rather than as definitive evidence to refute the conclusions of MARS2. This is a critical point that warrants clarification in the manuscript.
  3. If the MARS2 trial regulations included specific provisions regarding the secondary use of trial data—such as requirements for co-author consent or consensus within the research team—then it is essential to clarify whether this study was conducted in accordance with those regulations.
  4. Additionally, the authors should explicitly state whether this manuscript was shared with other MARS2 co-authors and whether permission or approval for publication was sought—or, if such approval was not required, provide a clear justification.
  5. Particularly, disclosing the responses from key investigators and co-authors of MARS2 would enhance the transparency of the study, allowing readers to assess the credibility of this analysis more accurately.

Author Response

Comment 1: 

  1. A critical ethical concern regarding this study is whether the reanalysis of MARS2 trial data falls within the scope of the original informed consent. If the initial consent was limited to analyses strictly within the trial framework, additional IRB approval and, if feasible, participant re-consent may be required. However, if the trial protocol explicitly permitted secondary analyses or follow-up investigations, this reanalysis may not pose significant ethical issues. Given the importance of this issue, I recommend that the Editor inquire with the authors regarding the ethical approval status and scope of informed consent for this study. To ensure clarity and transparency, please explicitly describe how these ethical aspects were considered and what procedures were undertaken to comply with ethical guidelines

Reply 1: We will enclose a copy of the patient consent form which states " I donate information collected as a gift which may be used for future ethical research"

Comment 2: 

  1. A key issue in this study is the verifiability of its conclusions. The authors argue that appropriate patient selection could demonstrate the effectiveness of surgery; however, verifying this claim would likely require a new prospective trial rather than a retrospective reanalysis of selected MARS2 cases. Since retrospective analyses inherently limit the ability to establish causality, it may be more appropriate to position this study as a hypothesis-generating investigation rather than one that provides definitive evidence. While presenting survival data using the 9th TNM staging is valuable, it should be explicitly stated that these findings serve as a basis for future trial designs rather than as definitive evidence to refute the conclusions of MARS2. This is a critical point that warrants clarification in the manuscript.

Reply 2: We agree with the reviewer and have clearly stated in our conclusions that this study merely acts a stimulus for further research into the role of surgery in mesothelioma rather than a justification to operate in an uncontrolled fashion.

Comment 3: 

  1. If the MARS2 trial regulations included specific provisions regarding the secondary use of trial data—such as requirements for co-author consent or consensus within the research team—then it is essential to clarify whether this study was conducted in accordance with those regulations.

Reply 3 : We do not believe that there are any restrictions on us a co-investigators in this respect.

Comments 4/5: 

  1. Additionally, the authors should explicitly state whether this manuscript was shared with other MARS2 co-authors and whether permission or approval for publication was sought—or, if such approval was not required, provide a clear justification.
  2. Particularly, disclosing the responses from key investigators and co-authors of MARS2 would enhance the transparency of the study, allowing readers to assess the credibility of this analysis more accurately,

Reply 4/5: This work has been presented several times to other authors of the MARS2 manuscript including the trial CI they are aware of and have discussed these findings in open forum. Of course we have not sought their permission, nor do we believe that ethically it is required. This work has been conducted independently of them but if the Editor requests so we are happy to add an acknowledgement of all of their efforts in the conduct of the main MARS2 trial. We do not accept that the "credibility" of this analysis is in anyway challenged by the absence of their specific consent. We would also assert that, as the majority of the trial surgeons, we are uniquely entitled to be considered as key investigators.

Reviewer 2 Report

Comments and Suggestions for Authors

As a reviewer I cannot declare anything else but express my deepest respect & gratitude towards the authors who are brave & scientifically / morally sound enough to stop a dangerous process: excluding surgery from the treatment of all stages/forms of malignant pleural mesothelioma. 

This is an exemplary protest against blind EBM fetishism; a call for critical reading/interpretation.

The paper is excellent: nothing to add / modify - congratulations.

Author Response

Comment : the authors who are brave & scientifically / morally sound enough to stop a dangerous process: excluding surgery from the treatment of all stages/forms of malignant pleural mesothelioma. 

This is an exemplary protest against blind EBM fetishism; a call for critical reading/interpretation.

Reply: we are very grateful for these compliments. We emphasize our belief in the importance of presenting our findings to challenge the misconceptions surrounding the MARS2 trial and the potential harm they may have on patient care. 

Reviewer 3 Report

Comments and Suggestions for Authors

Thank you for asking me to review the manuscript entitled "Why the MARS2 trial does not mean the end of all mesothelioma surgery".

This is an extremely interesting paper, discussing some aspects of the MARS2 trial that have been much debated on. Undoubtedly, the manuscript has the advantage of enlisting among the authors some of the main participants to the trial. The analysis and the results are interesting and well supported and I totally agree with the authors conclusions.

I have some considerations/suggestions:

- I found it quite difficult to follow the flow of patients to get to the final group. Starting from the abstract you say that from the initial 79 patients, you excluded 55, resulting in 24 patients (lines 25-29). However, group B is made of 8 cT1N0, 7cTN1N1 and 7cT2N0, which makes 22 not 24 (line 29) and in the main text the number is again different, because you considered 21 patients.

Again, in the main text (lines104-141) it is not very clear the patients’ flow. I think it would be better to make two different flow charts: one for the patients that actually met the preoperative criteria (21) the other for the ones that were upstaged or had a change in histology according to the final pathological report (to get to the final 12 patients that you report).

- I understand that the upcoming 9th TNM edition would re-classify some patients in the MARS2 trail, and actually help to better select patients. However, I think that is not quite fair to apply some modern criteria to patients that were selected/staged according to the classification used during the trial. At least I think it would be more correct to make two different discussions. One based on the classification that was actually used, the second implementing the discussion considering the upcoming changes in mesothelioma staging. I think this could make things more fair and valuable.    

- Survival (lines 143-144): could you add the survival among the other 58 patients as a matter of comparison?

- Line 130: I think you meant epithelioid mesothelioma not non-epithelioid.

- Line 140: I would say on final pathology not histology.   

Author Response

Comment 1: - I found it quite difficult to follow the flow of patients to get to the final group. Starting from the abstract you say that from the initial 79 patients, you excluded 55, resulting in 24 patients (lines 25-29). However, group B is made of 8 cT1N0, 7cTN1N1 and 7cT2N0, which makes 22 not 24 (line 29) and in the main text the number is again different, because you considered 21 patients.

Reply 1: We have reanalyzed the data and recalculated the results to correct the above mistakes.

Comment 2: Again, in the main text (lines104-141) it is not very clear the patients’ flow. I think it would be better to make two different flow charts: one for the patients that actually met the preoperative criteria (21) the other for the ones that were upstaged or had a change in histology according to the final pathological report (to get to the final 12 patients that you report).

Reply 2: As requested we have simplified the results section and identified the two patient flows as requested.

Comment 3: - I understand that the upcoming 9th TNM edition would re-classify some patients in the MARS2 trail, and actually help to better select patients. However, I think that is not quite fair to apply some modern criteria to patients that were selected/staged according to the classification used during the trial. At least I think it would be more correct to make two different discussions. One based on the classification that was actually used, the second implementing the discussion considering the upcoming changes in mesothelioma staging. I think this could make things more fair and valuable.   

Reply 3: The application of modern selection criteria in retrospect to the MARS2 patients is exactly the point of the paper. If clinicians are happy to apply the conclusions of the MARS2 trial to contemporary practice then surely it is fair to apply contemporary criteria to the MARS2 cohort.

Comment 4: - Survival (lines 143-144): could you add the survival among the other 58 patients as a matter of comparison?

Reply 4: As part of the reanalysis the comparative survival values have been calculated and presented.

Comments 5/6: 

Line 130: I think you meant epithelioid mesothelioma not non-epithelioid.

  • Line 140: I would say on final pathology not histology.  

Reply 5/6: These minor corrections have been made.

Round 2

Reviewer 3 Report

Comments and Suggestions for Authors

Thank you for the corrections you made to the manuscript that fullfilled my previuous comments.

Just one minor point: 

- Line 41: cT/4 should be cT3/4